# An Insulated Flexible Sensor for Stable Electromyography Detection: Application to Prosthesis Control

**DOI:** 10.3390/s19040961

**Published:** 2019-02-24

**Authors:** Theresa Roland, Kerstin Wimberger, Sebastian Amsuess, Michael Friedrich Russold, Werner Baumgartner

**Affiliations:** 1Institute of Biomedical Mechatronics, Johannes Kepler University, 4040 Linz, Austria; kerstin1995@gmx.at (K.W.); werner.baumgartner@jku.at (W.B.); 2Research and Development, Otto Bock Healthcare Products GmbH, 1110 Vienna, Austria; sebastian.amsuess@ottobock.com (S.A.); michael.russold@ottobock.com (M.F.R.)

**Keywords:** insulated sensing, capacitive sensing, electromyography, biosignal, flexible sensor, textile sensor, active sensor, upper-limb prostheses

## Abstract

Electromyography (EMG), the measurement of electrical muscle activity, is used in a variety of applications, including myoelectric upper-limb prostheses, which help amputees to regain independence and a higher quality of life. The state-of-the-art sensors in prostheses have a conductive connection to the skin and are therefore sensitive to sweat and require preparation of the skin. They are applied with some pressure to ensure a conductive connection, which may result in pressure marks and can be problematic for patients with circulatory disorders, who constitute a major group of amputees. Due to their insulating layer between skin and sensor area, capacitive sensors are insensitive to the skin condition, they require neither conductive connection to the skin nor electrolytic paste or skin preparation. Here, we describe a highly stable, low-power capacitive EMG measurement set-up that is suitable for real-world application. Various flexible multi-layer sensor set-ups made of copper and insulating foils, flex print and textiles were compared. These flexible sensor set-ups adapt to the anatomy of the human forearm, therefore they provide high wearing comfort and ensure stability against motion artifacts. The influence of the materials used in the sensor set-up on the magnitude of the coupled signal was demonstrated based on both theoretical analysis and measurement.The amplifier circuit was optimized for high signal quality, low power consumption and mobile application. Different shielding and guarding concepts were compared, leading to high SNR.

## 1. Introduction

Upper-limb prostheses are extremely helpful devices for people with amputations. By using surface electromyography (sEMG), so-called myoelectric prostheses can move actively. There are already various sEMG electrodes available on the market, and numerous research articles have focused on pattern recognition to enable movement at a high level of dexterity [1,2]. However, due to unsatisfactory robustness against various interferences [3], most of these systems have not yet been applied for daily use. The amputee must choose between a basic system with low functionality and a system that enables high dexterity but requires more cognitive effort [4]. If the system is unstable or requires excessive cognitive effort, amputees might switch to passive use or even reject the prostheses entirely, which leads to detrimental effects on the contralateral limb due to overuse [5]. Robust control is highly important to result in great acceptance of the myoelectric prosthesis [6]. The level of dexterity must be balanced such that the prosthesis can become part of the patient’s body [7].

Most research into prostheses and exoskeleton control has concentrated on state-of-the-art sEMG electrodes, which need a conductive connection to the skin [8,9,10]. These electrodes have various limitations due to the effects of skin perspiration, hair and fat content in the skin, and may cause skin irritations. Amongst others, these limitations lead to reduced stability of the sEMG electrodes. Hence, Cho et al. [11] and Radmand et al. [6] suggested using force myography as an alternative to sEMG. Connan et al. [3] proposed multimodal sensing to increase the robustness and reliability of myocontrol. They combined force- and electromyography in a device for myocontrol. Fougner et al. [12] combined EMG and accelerometers for hand motion classification.

Rather than alternatives or multimodal sensing approaches, this paper presents a capacitive EMG sensor that overcomes the limitations associated with state-of-the-art conductive sEMG electrodes. Capacitive sensors, also known as insulated electrodes, are insulated against direct contact to the skin. The electrolytic layer that is required in conductive electrodes is formed by an electrolytic gel or by a sweat film. In capacitive electrodes, this layer is replaced with a dielectric film. Due to the electrode’s design, no ohmic contact exists, so only displacement currents occur. Furthermore, no polarization effects arise, as there is no net charge flow from skin to electrode. Another advantage is that these sensors can be applied without skin preparation. A bio-compatible sensor design is straightforward, as only the insulating layer is in contact with the skin. Prance [13] compared wet, dry and active insulated electrodes for biopotential sensing. She pointed out the problem of movement artifacts, which occur with all electrodes.

Unlike for conductive sEMG sensors, little research has been published on capacitive biopotential sensors. Since electrocardiography (ECG) has higher signal amplitudes than EMG, most insulated biosignal sensors have been developed for ECG [14,15,16]. The idea of measuring biopotentials with capacitive sensors is old, but continues to offer great potential. As early as 1967, Richardson et al. [17] patented an electrocardiography and bioelectric capacitive electrode using a transistor circuit for impedance conversion. In recent research, several systems have been suggested for capacitive biopotential sensing. These sensors have been developed for various applications, such as prosthesis control and exoskeletons, and use different techniques to deal with bias currents at the input. Various methods have been proposed for referencing to the body to reduce the subject’s common-mode voltage. Shielding is also dealt with, as it plays an important role in insulated biopotential electrodes.

Insulated electrodes allow biopotentials to be measured even through thin cloth or across a small air gap. Lim et al. [16] presented a system for measuring the ECG signals of a subject who is sitting on a chair and wearing clothes. This system requires a large ground plate integrated into a seat surface of the chair. Clearly, integration of such a large ground plate is impossible in other applications such as upper-limb prostheses. Lee et al. [14] also developed a capacitive sensor for measuring ECG signals through cloth, where the active electrode was designed to be thin and flexible. They implemented an active shield to reduce stray capacities. A capacitive driven-right-leg electrode with maximized surface area on the chest belt was used to reduce the subject’s common-mode voltage. In their work, motion artifacts distorted the signals; these artifacts limit biopotential measurement. As long as the signal is in the operating range of the analog circuit, the signal peaks can be identified after bandpass filtering. Further, Spinelli et al. [15] and Ueno et al. [18] measured biopotentials through cloth. The former [15] provided a detailed description of the input circuit and the latter [18] showed ECG and EMG measurement through cloth. Insulated sensors have even been used in underwater applications [19]. Oehler [20] developed a capacitive ECG sensor and a capacitive electroencephalography (EEG) helmet in his dissertation. Loong et al. [21] compared a variety of textile materials used as insulators for biopotential sensing. However, the sensing area itself was not textile but a rigid copper plate.

Various ways of applying insulated biopotential sensors have been described in the literature. They can be used, for example, for ECG, EEG and EMG. Here, we present sensors developed for capacitive EMG measurement. They can be employed in basic systems which allow simple movements of a prosthesis, in an array enabling a high dexterity level, or even in combination with other measuring methods. We focused on an application for a standard upper-limb prostheses available on the market and provided by Otto Bock Healthcare GmbH. This standard prosthesis uses EMG sensors at two muscle groups. By co-contraction, the movement mode of this prosthesis can be switched; this is achieved by simultaneous short, strong contraction of both muscle groups. Clearly, this sensor can be adjusted to other applications.

Our insulated sensor is highly robust and designed for clinical application. Various sensor assemblies were investigated for optimal signal coupling, using flexible materials to achieve good adaption to the anatomy of the human forearm. The resulting measurement system is compact, but well thought out. It aims for ease of use, long-term application, low-noise performance and stability and usability in a real-world environment. We presented a previous prototype of our measurement set-up [22] and its digital signal processing path [23].

First, we present the fundamentals of the conductive and capacitive sensing principles. We then describe various different sensor assemblies and the sensor electronics of our capacitive EMG measurement system. This is followed by a presentation of the transfer functions of the measurement set-up. We then compare the results from experiments using the different sensor assemblies and describe the shielding set-ups.

## 2. Comparison of the Capacitive and Conductive Measurement Principles

To explain the challenges associated with capacitive EMG measurement, we briefly compare the capacitive and conductive measurement principles in this section. Figure 1 shows the principle of the capacitive and conductive measurement set-up with the coupled amplitudes. The transmission of the action potential to the human skin via the electrical network in the human tissue was covered by Roland et al. [24].

The input amplifier stage is realized by an instrumentation amplifier (INA) with high common-mode rejection ratio (CMRR) for interference reduction. The main difference in terms of INA between conductive and capacitive measurement is that the direct current (DC) operating point is defined by the reference electrode in the former case and by the bias resistor *R_B_* in the latter case.

A stable DC operating point within the operating range of the amplifier is essential to signal acquisition. The measurement signal is transmitted via alternating current (AC) transmission behavior.

For a stable DC operating point, capacitive measurement requires a bias resistor *R_B_* through which the bias current can flow. The capacitive measurement principle forms a highpass, which attenuates the low-frequency components. Sensing of biopotentials is sensitive to low-frequency interferences, which are already attenuated due to the highpass resulting from the measurement principle.

The signal is not only attenuated in the low-frequency components, but across the entire frequency spectrum. The smaller signal amplitude at insulated electrodes results from the impedances at the voltage divider, which is formed by *C_C_* and the impedance of *C_P_* parallel to *R_B_*. The adjustment of these impedances is explained in Section 3.1.1.

Conductive measurement requires no bias resistor, because the bias current path is established via the conductive connection to the skin. Therefore, the high input impedance of the amplifier can be exploited. The DC potential of the signal lines are defined by the reference electrode, which also has a conductive connection to the skin.

Despite the smaller amplitude, the capacitively coupled EMG signal can be measured with high-precision electronics.

## 3. Sensor System Description

A block diagram outlining the measurement system is shown in Figure 2. These blocks are described in the following sections.

The capacitive EMG measurement system must fulfill the following requirements:stability in real-world environment;low power consumption;low cost;compact size;ease of use;high wearing comfort;real-time capability; andhigh signal-to-noise ratio (SNR).

### 3.1. Interface to Human Body

The capacitive EMG sensing electrode and the body reference form the interface to the human body. These are described in the following paragraphs.

#### 3.1.1. Capacitive EMG Sensing Electrode

The sensing electrode couples the EMG signal from the skin to the electronic circuit. Based on previous evaluations [24], five capacitive and one conductive assemblies were compared. Hence, we designed various EMG sensors that are multilayer constructs consisting of various materials to compare their signal coupling behavior. The sensors in this work were fabricated by stacking flexible materials such that they can adapt to the human forearm anatomy (Figure 3a). The air gap between skin and electrode was minimized to achieve better signal coupling. The flexible materials used were copper foil stacked with plastic foil (foil sensors), multilayer flex print (flex sensors) and stacked conductive and insulating textiles (textile sensors).

The structure of such a sensor is shown in Figure 3b. This sensor measures in differential mode, so one module consists of two symmetrical sensor areas. A shield covers the sensor areas and the signal wires to increase common-mode rejection and to protect against interferences. For the textile and foil sensors, the shielding of the wire was realized by means of ultra-miniature coaxial cables (UMCC) [25] with 1.37 mm outer diameter. The flex sensors are a multi-layer flex print: one layer serves as sensor area and the other is used for shielding. They are directly connected to the electronics, so no separate wires are required.

The six different sensor assemblies (Figure 3c–e) have the parameters as listed in Table 1 and the following paragraphs provide supplementary information.

Sensor 1 has two layers of adhesive, with 30 μm thickness each, which are positioned between the dielectric and the conductive layer: one at the dielectric and one at the conductive layer. The technical characteristics of the adhesive layers are not stated in the data sheets, so the values listed by Dupont^®^ [26] for their acrylic adhesives were used in the assumptions. The conductive layers are formed by self-adhesive copper tape [37].

Sensor 2 is made of dielectric and conductive textile layers, which were pressed for bonding. The UMCC cables are held in place by silicone. The silicone layers between skin and sensor area and between sensor area and shield were estimated to be 100 μm. Note that the silicone was applied when liquid, so the thicknesses listed are estimations and not exact values. The parameters of the silicone were estimated by measuring the capacity of the corresponding layers.

Sensor 3 is also a pressed textile multi-layer construct. The silicone layers, which fix the UMCC cables, were estimated to be 100 μm thick.

The adhesive layer thickness is <10% of the overall thickness in the flex prints (Sensors 4–6) [27]. Thus, the adhesive in the flex prints is neglected.

Sensor 6 is a conductive sensor that resembles Sensors 4 and 5, but without a dielectric that covers the conductive sensor areas and without electrolytic gel.

#### 3.1.2. Body Reference

Common-mode voltage components are caused by the mains frequency (50 Hz in Europe and 60 Hz in North America), by movement artifacts, by static charges or by other interferences. The sensor electronics must be connected to the body such that the common-mode component remains within the operating range of the amplifier. If the signal is within the operating range, the amplifier common-mode suppression works and the AC signal can be measured.

Oehler [20] used a wet electrode as a low-impedance path to the body. Thus, accumulation of electrostatic charges was prevented [38]. Lim et al. [16] referenced to the body with a large-area plate capacitor at the seat surface of a chair. Haberman et al. [39] presented a fully capacitively driven-right-leg concept based on a common-mode negative feedback.

In our case, using a large-area capacitive electrode is not practicable for a small system. Since the driven-right-leg system is less stable than the conductive reference, we chose a dry conductive reference that can be placed directly on the forearm without skin preparation or applying an electrolyte. This reference electrode was realized by means of a conductive textile sewed into a cuff, which was worn at the forearm in the measurements, as shown in the measurement set-up in Section 5. A wet electrode would not have allowed long-term monitoring. Using a dry conductive reference benefits the system, as the reference electrode is not as sensitive to lifting, sweating or hair as a conductive EMG sensing electrode. The advantages of the capacitive biopotential measurement are therefore retained. A small reference is sufficient and can be placed anywhere on the forearm. Placement and size are interdependent, and skin condition also affects the minimum size required. A reference electrodes with a size of 1–2 cm^2^ was sufficient for a working set-up. However, this work includes no quantitative analysis of the reference electrode size and positioning. When using two or more EMG sensor systems, only one reference is required.

#### 3.1.3. Connection to Circuit Board

The foil (Sensor 1) and textile (Sensors 2 and 3) sensors are connected to the circuit board via UMCC cables and a UMCC plug. The shield surrounds the EMG signal line from the plug to where the EMG signal enters the INA, where it is impedance-converted and amplified. The EMG signal lines are as short as possible to prevent stray capacities and interferences from coupling to the signal. The flex sensors (Sensors 4–6) are connected to the circuit board via a flat band plug. The flex sensor material at the plug connection is strengthened to achieve a mechanically stable connection. The flat band plug is mounted on the layer of the PCB print opposite to the UMCC plugs to minimize the signal path and have the shield surrounding the EMG signal line.

### 3.2. Analog Circuit

The final circuit diagram for the insulated EMG sensor electronics is shown in Figure 4. In the following sections, these blocks are described in more detail, and the choice of components is justified. The signal is shielded, preamplified and filtered by an analog bandpass. The following stage is the digital signal processing with a microcontroller. A bluetooth-low-energy (BLE) module on board establishes a bidirectional wireless connection to the PC in experimental applications and is used for setup of controller parameters as well as for measurement data transmission.

#### 3.2.1. Power Supply

The power supply circuit diagram is shown in Figure 4 (top left). The sensor is designed for power supplies from 5.3 V to 16 V, which is the typical voltage range of upper-limb prostheses batteries. For the capacitive EMG measurement electronics, a supply voltage of 3.3 V is generated. The power is supplied by an Otto Bock upper-limb prosthesis battery with 7 V. This battery also supplies the prosthesis drive. Start-up currents at the prosthesis drive cause the 7 V power line potential to drop by ≈0.5 V. Due to pulse width modulation of the drive, significant spikes with high frequency components interfere with the supply voltage. Further, the bluetooth module causes spikes in the supply when transmission is activated. Measuring EMG with insulated sensors requires precision electronics with highly stable power supply, so an appropriate choice of voltage regulators is crucial. Blocking capacitors *C_S_* are used to stabilize the power supply. They are placed as close as possible at each component to minimize power-line impedances and achieve better stabilization effects. For better visibility, some *C_S_* are not shown in the resulting circuit diagram in the figure.

The power supply consists of two stages of voltage regulators, which were selected primarily for their power supply rejection ratio (PSRR) and their line transient response. The first stage attenuates the voltage drops and spikes of the prosthesis supply and provides a stable 5 V potential. The LD2985BM50R with a PSRR of 45 dB at 1 kHz and an excellent line transient suppression is used in the first stage [40]. A bypass capacitor *C_BP_* with 10 nF is connected between the bypass pin and ground, as suggested by the data sheet.

The second voltage regulator AP7312 [41] generates two supply voltages at 3.3 V: one is connected to the bluetooth module, and the other to the analog circuit and the microcontroller. The μC connects analog and digital supply voltage internally. However, measurements showed that the controller does not cause spikes in the supply line. This voltage regulator was chosen because it provides two output voltages and has a good line transient suppression; the PSRR is 65 dB at 1 kHz.

#### 3.2.2. Instrumentation Amplifier (INA)

The AD8236 [42] instrumentation amplifier (INA) and its wiring are shown in the *Circuit Input* block in Figure 4 (middle left). The INA must fulfill the following requirements:low bias current (limits *R_B_*);pin assignment that enables shielding;supply voltage of 3.3 V;rail-to-rail input and output;high CMRR;high PSRR;low noise level;low power consumption; andlow cost.

The selected INA, the AD8236, fulfills these requirements. The potential level at the input stays within the operating range, due to the low bias current in combination with the selected *R_B_*. The INA is low-power, as it uses a maximum of 40 μA supply current. The rail-to-rail input and output and the high CMRR are necessary for the measurement system. The operating range fits the 3.3 V power supply. The AD8236 provides the signals at the negative inputs of the internal operational amplifiers (OpAmps) at pins 2 and 3. Its potential corresponds to the input signal, which enables shielding. The gain resistor *R_G_* is placed between those pins. *R_G_* is split into two resistors (*R_G1_* and *R_G2_*) in series with 10 kΩ each, resulting in a gain of 26 at the first stage, according to the gain equation:(1)$$G=\frac{420\;k\Omega }{R_{G}}+5.$$

*R_IP_* protects the amplifier input pins from high currents at electrostatic discharges. *C_AC_* is used to have AC coupling if a conductive sensor is applied.

##### 3.2.2.1. Bias Resistor

The bias resistor *R_B_* was chosen to be 100 MΩ, which is relatively low compared to the values in the literature. Lee et al. [14] used an *R_B_* with 5 GΩ, and Sullivan et al. [43] employed a reset circuit with two transistors to bring the input signal back to the amplifier’s operating range. Ohtsu et al. [19] did not provide any information about the drift caused by bias current. Very high input impedances must be realized with care: If the circuit board is contaminated, for instance, by a fingerprint or some other particles, the resistance between the signal lines might be smaller than intended.

The value of *R_B_* was determined empirically. It was chosen to be 100 MΩ to achieve a stable DC operating point at the INA. The bias current *I_B_* of the AD8236 amplifier is typically 1 pA and at most 10 pA [42], which results in a potential shift relative to the *V_REF_* of typically 0.1 mV and at most 1 mV. The selection of *R_B_* is a trade-off between stability and bandwidth. Incorporation of the bias resistor into the final circuit is shown in Figure 4.

The resistor *R_B_* affects the cutoff frequency of the input, as described in Section 4.2 and Section 6.2.1. The EMG signal power is in the frequency range from 0 to 400 Hz [44]. Biopotential measurement is sensitive to low-frequency artifacts. This particular measurement set-up attenuates the movement artifacts, but at the cost of the EMG signal power in the low-frequency range also being attenuated. Nevertheless, a low *R_B_* was chosen deliberately because it leads to higher stability of the system.

The low impedance towards ground also reduces the magnitude of the coupled interferences. The lower signal amplitude caused by a lower *R_B_* can be compensated for by a higher amplifier gain. At higher *R_B_*, the gain is limited in any case to keeping the signal within the AC operating range.

##### 3.2.2.2. Reference Potential

The INA requires a reference voltage *V_REF_* at a potential between positive and negative supply. Since this insulated EMG sensor uses a single supply, the negative supply is the ground potential and *V_REF_* must be generated. Two 68 kΩ resistors form a symmetric voltage divider between positive supply and ground. This *V_REF_* potential of 1.65 V is impedance-converted at the MCP6022’s [45] second OpAmp and connected to the AD8236 [42] *REF_PIN_*.

##### 3.2.2.3. Alternative Reference Design: INA with DC Rejection Reference Design

INA with DC rejection reference design can be used as an alternative to a fixed reference potential at the INA. An integrator feeds back the AD8236 output to the *REF_PIN_*. This electrical circuit acts as a highpass filter directly at the first amplification stage. AC differential input signals are amplified, and DC differential and common-mode signals are rejected [46]. The AD8236 data sheet outlines an application with this circuit [42]. A highpass with a cutoff frequency depending on RC with
(2)$$f_{C}=\frac{1}{2\pi RC}.$$
is implemented by means of a feedback to the reference of the INA (see Figure 5). The fed back reference potential floats with the inverted low-frequency interferences, thus eliminating them. Filtering the interferences only makes sense if the signal is within the operating range. For this reason, a highpass at the first amplification stage would be highly beneficial, because signal saturation due to movement artifacts might already occur at the INA.

We built electric circuits with and without DC rejection reference design and compared them.

#### 3.2.3. Shielding/Guarding

Shielding protects the input signal from external interferences. Guarding is achieved by a low impedance conductor which surrounds the input signal line and prevents parasitic capacities. In the circuit we implemented, these tasks are fulfilled by the common-mode shield or active shield. We hereafter subsume both shielding and guarding under the term shielding.

Various methods for shielding the input signal lines from external interferences and preserving signal quality until the signal enters the INA have been presented in the literature. Jiang et al. [47] compared no shielding as well as active-, ground- and bias-shielding methods for biopotential signal acquisition. They found that active shielding led to the best 50 Hz hum suppression. Bias shielding exhibited poorer hum suppression than active shielding but performed better than ground shielding. Nevertheless, ground shielding was better than no shielding: it protects against external interferences, but causes parasitic capacities [48].

In theory, active shielding and common-mode shielding are the most promising methods for the capacitively coupled EMG signal (see the following paragraphs). Hence, we implemented these two shielding methods in a physical circuit for comparison.

##### Active Shielding

The most common shielding method applied in insulated biopotential measurement is active shielding, where each input signal is impedance-converted and connected to the shield. Maintaining the same AC voltage between signal line and shield leads to compensation for parasitic capacities in the EMG signal [49]. Each operational amplifier, used to generate the active shield potential, causes noise, which is coupled to the signal line [50]. Since this is a differential mode noise, as each input signal line has its own operational amplifier to generate the active shield potential, it is amplified in the signal processing chain.

The active shield set-up we implemented in a physical circuit consists of an AD8236 instrumentation amplifier and an MCP6022 Dual OpAmp for the voltage followers (see Figure 6). The AC potential at the gain resistor pins of the INA is equivalent to the input signal. The potentials at both gain resistor pins are impedance-converted at the voltage followers and connected to the sensor shield. Due to capacitive coupling of the shield to the signal line, a different DC potential at the gain resistor pins, resulting from the internal circuit of the instrumentation amplifier, does not affect the shielding. The gain resistors *R_G1_* and *R_G2_* are 10 kΩ.

Another option for the active shielding circuit would be using an INA which internally generates the active shield potential and provides it at a pin. For INAs that do not provide the active shield or input signal potential, a voltage follower could be used for impedance conversion at a stage before the INA, but this would lead to a higher component count and poorer noise performance.

##### Common-Mode Shielding

The shield is driven to the same common-mode potential as the input; this way the leakage current is eliminated for common-mode potential [48]. Common-mode shielding causes parasitic capacities for differential-mode signals, but it greatly increases the common-mode rejection by minimizing the common-mode impedance, compensates for the effect of unequal cable capacities and minimizes leakage current [49,51].

The common-mode shielding we implemented is shown in Figure 7. The common-mode shielding potential is tapped from the central point of the two series gain resistors *R_G1_* and *R_G2_*, with 10 kΩ each, at the AD8236 [42]. This signal is connected to the input of the voltage follower at one of the MCP6022 Dual OpAmps [45].

#### 3.2.4. Analog Bandpass

The INA AD8236 [42] output is filtered by a passive first-order lowpass with 1064 Hz cutoff frequency. The lowpass is formed by *R_LP_* and *C_LP_* (Figure 4). The high-frequency components are eliminated to attenuate interferences and to prevent aliasing effects when digitally sampling the signal. The lowpass is followed by a passive first-order highpass with 11 Hz cutoff frequency (see component *C_HP_* and *R_HP_* in Figure 4). The signal is DC-decoupled by this highpass, so the required DC potential is set by the microcontroller DAC (see Section 3.2.5). The highpass attenuates the low-frequency interferences, particularly the motion artifacts, which commonly occur in biopotential measurement. The two passive filters are cascaded directly without a voltage follower in between. By selecting higher component values at the second stage, the lowpass, the passband attenuation remains small. The cutoff frequencies are chosen to maintain the EMG signal power while filtering low- and high-frequency interferences. The resistor *R_HP_* of the highpass is connected to one of the microcontroller DACs (see Section 3.2.5 and Figure 4).

#### 3.2.5. Digital Signal Processing (DSP)

The ultra-low-power microcontroller (μC) ATSAML21E18B [52] is used for digital signal processing.

##### Microcontroller Input

The μC DAC defines the DC potential of the μC input signal, which is DC-decoupled at the previous analog lowpass at *C_HP_* (see Figure 4). The internal OpAmps of the μC amplify the signal by an adjustable gain. The combination of the three OpAmps depends on the selected gain. As they amplify the DC component, the potential at the DAC must be set to a value determined by the selected gain. The DC level at the ADC is set by the offset at the DAC and the OpAmp gain. In the described set-up, gains in the range from 8 to 64 are usually applied, depending on user and application. A gain of 8 implies a 206 mV DC offset, which is determined by the DAC, so the signal DC component is 1.65 V after amplification. The DC level after amplification must be half the supply voltage to use the full operating range of the analog-to-digital converter (ADC), which is from supply voltage to ground. The gain is configured by the microcontroller software by setting the OpAmp interconnections and resistor values.

The output of the last OpAmp is connected to the microcontroller ADC, the reference voltage of which is set to supply voltage. The microcontroller [52] supports on-board accumulation and averaging functionality at the ADC. Accumulation leads to a higher resolution at the ADC, and averaging improves the noise performance. Sixty-four samples at the ADC are accumulated and averaged at each sampling period. Sixteen samples are accumulated to increase precision and four samples are averaged to improve the noise performance. The 12-bit precision of the ADC increases to 16-bit precision even at higher noise performance. The sampling rate for the EMG signal is 10 kHz.

##### Digital Filtering

For the measurements, a second-order digital highpass filter with a cutoff frequency of 60 Hz was implemented to eliminate low-frequency interferences. A second-order digital notch was implemented to filter the 50 Hz hum and its harmonics. These filters were designed to use minimal calculation resources for low-power applications. The digital filters were described in detail by Roland et al. [53].

#### 3.2.6. Alternative Approaches

Alternative approaches to improving EMG signal coupling and the reference to the human body were considered for the analog circuit, but these were not included in the final circuit.

##### Bootstrapping and Neutralization

Bootstrapping (Figure 8a) is another method of providing a stable DC operating point at the amplifier input. A positive feedback via a capacitor increased the apparent input impedance relative to the signal frequency. At frequencies above the cutoff frequency, defined by *R*_*B*1_ and *C_B_*, the input impedance ideally appears to be infinite. As the capacitor does not feed back DC signals, the bias currents can flow via the resistors, preventing the signal from saturating. Portelli et al. [54] and Prutchi et al. [55] applied bootstrapping in their circuits.

Using a neutralization circuit allows compensating for the parasitic and input capacities *C_P_* and *C_IN_*, as employed, for instance, by Spinelli et al. [15]. This is realized by means of a positive feedback with a gain ≥1. A current is provided that equates the current flowing in *C_IN_* and *C_P_* to compensate for these capacities. The equivalent circuit diagram for neutralization is shown in Figure 8b. Heuer [56] implemented both the neutralization and the bootstrapping circuit into an ECG electrode.

However, we found that neutralization results in poorer noise performance, which corresponds with Fein [57]. The bootstrapping and neutralization circuits lead to an increased component count and greater size and cost of the circuit. Bootstrapping and neutralization increase the highly sensitive high-impedance EMG signal line’s length. The positive feedback loops are connected directly to the EMG signal line and may introduce additional interferences to the sensitive EMG signal. Noise is introduced due to the feedback and the OpAmp at neutralization. Since positive feedbacks involve the risk of instabilities and bootstrapping and neutralization do not meet the low-noise and low-component requirements, they were not included in the final insulating EMG measurement circuit.

##### Driven-Right-Leg

The driven-right-leg approach, which we did not use in our set-up, is an alternative method of providing the reference to the human body. In a driven-right-leg set-up, the common-mode component is inverted and fed back to the patient. The selection of the feedback amplification is crucial, because this feedback loop is large and system stability is essential. In the driven-right-leg approach, additional electric components are necessary for shielding the input signal line. It is not suitable for multi-sensor applications, because the common-mode feedback voltage is driven by each sensor. Although this method is effective in reducing common-mode interferences, it affects differential-mode interferences in an unpredictable way and can increase interference [58]. Due to these limitations, we deemed this approach unsuitable.

## 4. Theoretical Analysis of the Measurement System

The theoretical approach was compared to the real-world measurements. The theoretical amplitudes were calculated with the transfer functions for the various insulated EMG sensors.

### 4.1. Input Signal

As an input for the theoretical analysis, the EMG signal at the skin surface was modeled. The use of real EMG data as an input was avoided, because any measurement already modifies the signal characteristics. When modeling the input signal, the stochastic time domain signal density was assumed to be Gaussian [59,60] and the peak-to-peak amplitude was set to 5 mV [59,61] (see Figure 9a). In accordance with the power spectra in literature [10,62], the shape of the amplitude spectrum was defined as plotted in Figure 9b. The smoothed amplitude spectrum was applied as input to the theoretical analysis of the measurement setup.

### 4.2. Input Stage

The input stage is the signal chain from the human signal source (stratum corneum) of the skin to the input of the INA, including the INA input impedance. The equivalent circuit diagram of the input stage is shown in Figure 10, and the transfer function is defined by:(3)$$G_{InputStage}\left(j\omega \right)=\frac{{\hat{V}}_{AD8236IN}}{{\hat{V}}_{Skin}}=\frac{Z_{2}Z_{4}}{\left(Z_{3}+Z_{4}\right)\left(Z_{1}+Z_{2}\right)+\left(Z_{1}Z_{2}\right)}$$
with the impedances
(4a)$$\begin{array}{c} \begin{array}{c} Z_{1}\left(j\omega \right)=R_{SC}+\frac{1}{j\omega C_{C}}, \end{array} \end{array}$$
(4b)$$\begin{array}{c} \begin{array}{c} Z_{2}\left(j\omega \right)=\frac{1}{j\omega C_{P}}, \end{array} \end{array}$$
(4c)$$\begin{array}{c} \begin{array}{c} Z_{3}\left(j\omega \right)=\frac{1}{j\omega C_{AC}}+R_{IP}, \end{array} \end{array}$$
(4d)$$\begin{array}{c} \begin{array}{c} Z_{4}\left(j\omega \right)=\frac{1}{\frac{1}{R_{B}}+j\omega C_{IN}}. \end{array} \end{array}$$

The plots of the transfer function are shown in Section 6.2.1. *R_SC_* is the impedance of the untreated stratum corneum, which was assumed as an ohmic resistance of 3 MΩ [63,64]. The coupling capacity *C_C_* is the capacity between skin and sensor electrode. *C_C_* was determined with the equation for a plate capacitor
(5)$$C_{Plate}=\varepsilon_{0}\varepsilon_{R}\frac{A}{d}$$
for each sensor with the associated dielectric (vacuum permittivity ε_*0*_, dielectric constant ε_*R*_, sensor area *A*, thickness *d*), as listed in Section 3.1.1. The resulting capacity *C_C_* can be seen in Table 2. For the sensors with adhesive tape or silicone fixation, the capacity *C_C_* was calculated as a serial connection of the capacitors formed by the dielectric layers. In Sensor 6, which is the conductive sensor, *C_C_* is a short circuit. The stacking of the sensor assemblies is illustrated in detail in Figure 11.

The capacity *C_AC_* with 10 nF AC couples the measurement signal, even when conductive electrodes are connected. The resistor *R_IP_* with 47 kΩ protects the INA input pins as it limits the current in case of static discharge. The bias resistor *R_B_* is 100 MΩ. The internal input capacity *C_IN_* of the INA is 3.1 pF [42].

The shield with the common-mode component is a parasitic capacity *C_P_* for the differential mode signal, which was calculated with *C_P_* connected to ground. The shield also covers the signal line connecting the sensor area to the electronics. At the flex sensors, this capacity *C_P_* of the shield is also calculated with Equation (Equation 5). For the UMCC cables the equation for the cylindric capacity
(6)$$C_{Cylinder}=2\pi \varepsilon_{0}\varepsilon_{R}\frac{l}{ln(\frac{r_{outer}}{r_{inner}})}$$
is used. The parameters for the parasitic capacity of the shield at the connection to the circuit board are listed in Table 3.

The capacity of the sensor area shield and the connection are in parallel and the resulting parasitic capacities *C_P_* are listed in Table 4.

### 4.3. Analog Bandpass and Digital Signal Processing

The electrical circuit and the component values for the transfer function of the analog bandpass can be seen in Figure 4.
(7)$$G_{AnalogBandPass}\left(j\omega \right)=\frac{{\hat{V}}_{\mu C,In}}{{\hat{V}}_{INA,Out}}=\frac{\frac{1}{j\omega C_{LP}}R_{HP}}{\left(\frac{1}{j\omega C_{HP}}+R_{HP}\right)\left(R_{LP}+\frac{1}{j\omega C_{LP}}\right)+\left(R_{LP}\frac{1}{j\omega C_{LP}}\right)}$$

The transfer functions for the digital signal processing include the second-order highpass with a cutoff frequency of 60 Hz and a 50 Hz second-order notch. The frequency independent amplification is also incorporated into the transfer function of the entire measurement system.

#### 4.3.1. Signal Delay

The signal delay was determined by the real-time capable controller algorithms, which Were described in detail by Roland et al. [53]. If the EMG signal was smoothed, the delay was dominated by the smoothing filter, a digital PT1-element with a T = 51.1 ms, which is not perceptible to the user.

## 5. Experimental

The same electronics were used in all measurements except for the components to be compared. The common-mode shielding set-up from Figure 7 was used for these comparisons. The comb- and highpass-filtered EMG signal was applied to the DAC and measured by means of a Handyscope HS3 oscilloscope [66].

In our set-up, the frequency independent gains were:26 at the INA;8 at the μC internal OpAmps;16 at the ADC (accumulation of 64 samples and division by 4);0.75 at the digital comb filter; and0.25 at the DAC (right-shift by two to be within the 12-bit value range of the DAC).

Note that the signal was damped in EMG frequency range by filtering and capacitive coupling. With this frequency independent amplification, this damping behavior was compensated. The frequency independent and frequency dependent gains depended on the sensor assembly used and they ranged from 342 to 1437 at the total signal chain at the peak frequency (see Section 6.2.3).

The Handyscope HS3 [66] was connected to the PC, and the Multi Channel software [67] from TiePie engineering was used to display the signals. The voltage range was set to ±4 V, the sampling frequency was set to 10 kHz, and the time per division was 1 s. Thus, we stored 10 s time snippets. The measurement ran for 10 s, but only data starting from 2 s were used in the evaluation, as the test subject was starting the contraction within the first 2 s. Therefore, distortion of the onset of the muscle contraction due to signal cancellation at low forces [68,69,70] and possible movement artifacts due to the relative movement of the muscle tissue to the EMG sensor were not included in the evaluation.

For the measurements, the EMG sensors were placed at the left and right human forearm. The distance between epicondylus lateralis and the ulna distal end was measured. At one third of this distance from the epicondylus lateralis, the sensor center was placed. The sensor was positioned along the longitudinal axis of the muscle, and a cuff was wrapped around the forearm for sensor fixation. It was crucial to place the sensor at exactly the same position in all measurements, as otherwise sensor positioning would have had a greater influence on the EMG signal power than the parameters to be compared (e.g., sensor assembly or shielding). The subjects performed a maximum voluntary static extension in the wrist joint by pulling back the hand against the restraint of the joint (see Jiralerspong et al. [71]). It was also highly important to apply the cuff with the same amount of pressure in each measurement. Figure 12 shows the measurement set-up (Figure 12a) and the PCB print (Figure 12b).

### 5.1. Normalization

The EMG power varied for different subjects, which had to be considered in the comparison. Hence, the sum of the root mean square (RMS) per person and per arm was normalized to 1 to compare sensor performance and not subject EMG power. For each subject, each RMS value was divided by the sum of the RMS values over all sensors according to:(8)$${\vec{V}}_{RMS}normalized=\frac{\left[V_{RMS}S1,V_{RMS}S2,\ldots ,V_{RMS}S6\right]}{\sum_{n=1}^{6}V_{RMS}Sn}$$
where $V_{RMS}S_{n}$ are the measured RMS values of Sensor 1–6.

### 5.2. Comparison of Sensor Assemblies

For optimal signal coupling at the interface between electrode and skin, various multilayer-sensor constructs were compared. The sensor descriptions can be found in Section 3.1.1. The sensor assembly was changed between measurements while the circuit board with the measurement electronics remained the same.

#### 5.2.1. Measurement Procedure

For this study, ten able-bodied subjects aged between 23 and 62 were selected. The subjects were asked to perform maximum voluntary muscle contraction of the musculus extensor digitorum. Three measurements were conducted per subject, arm and sensor, resulting in 18 measurements per forearm. Each measurement had a duration of 10 s, the contraction onset in the first 2 s was discarded and the remaining 8 s EMG of maximum voluntary wrist extension was used in the evaluation (see Section 5). There was a resting period of 30 s between the measurements and after every third measurement there was a longer resting period of 2 min, which was used to apply the next sensor to the forearm. The order of the sensors was randomized. Although the subjects were not blinded to the application of the different sensor assemblies, they were naive to the purpose of the experiment.

#### 5.2.2. Data Evaluation

The data evaluation was done in Matlab^®^ [72]. Eight seconds of actual EMG contraction were selected from the measurement files and the DC offset was subtracted. The three measurements of the same sensor, person and arm were stacked in an array. This stacking was performed for all measurements. One RMS value was calculated for each stack of three measurements.

The *V_RMS_normalized* for the left and the right arm were stacked into one array each, treating left and right arm as separate subjects. The mean and the standard deviation were calculated for each sensor based on the data of all subjects.

The resulting RMS values per sensor were compared statistically to determine which sensor coupled the highest EMG RMS. The RMS values for each sensor were tested for normal distribution by means of the Chi-Square Goodness of Fit Test [73]. To run multi-sample tests for equal variances, we employed the Bartlett Test [74]. Finally, we used pairwise comparison to test whether the sensor means were from the same group. The *t*-test [75] for equal means was applied with the according options. The significance levels were set to 5%.

The sensor noise level caused by the sensor assembly was negligible and deviations were caused by interferences from the environment rather than by the sensor assembly. Hence, we did not calculate the signal-to-noise ratios when comparing the sensors, as it would not have helped with answering the question of which sensor has the optimal EMG signal coupling.

#### 5.2.3. Fatigue Evaluation

To evaluate influence due to fatigue in the sensor comparison measurements, five able-bodied subjects aged between 26 and 52 were measured at the left and right forearm, resulting in 18 measurements at ten forearms. For the details of contraction and sensor placement, see Section 5. Sensor 5 was used for all measurements. There was a resting period of 30 s after each sensor and 2 min after every third measurement to resemble the sensor comparison measurements. In the sensor comparison measurements, the mean of three measurements was calculated per sensor. Correspondingly, the mean of three subsequent measurements was calculated in the fatigue evaluation. These values were normalized to eliminate inter subject deviations and, finally, the mean *V_RMS_normalized* of the five subjects was calculated, resulting in six mean *V_RMS_normalized* values. To evaluate the fatigue, the change of the *V_RMS_normalized* over the maximum voluntary contractions was evaluated. The subjects were naive to the purpose of the experiment.

### 5.3. Shielding (Active Shield and Common-Mode Shield)

Shielding is a key element of insulated biosignal sensing electronics. Active shielding and common-mode shielding were compared in terms of EMG signal coupling and noise performance. The electronics used in the measurements differed only in the shielding (Figure 6 and Figure 7).

#### 5.3.1. Shielding Measurement Procedure

For the comparison of the shielding circuits, four 26 year-old able-bodied subjects were measured at the left and at the right forearm. Three measurements per arm and subject were conducted at maximum voluntary contraction and one measurement was performed with the muscle relaxed to determine the noise. Note that the instrumentation amplifier input pins were not connected to *V_REF_* for this noise measurement to incorporate the shielding behavior at the input stage. These signals were measured with the active shield and the common-mode shield electronics. The copper sensor (Sensor 1) was used for these measurements. In the shielding measurements, the subjects were blinded to which electronics they were wearing and they were naive to the purpose of the experiment.

#### 5.3.2. Shielding-Data Evaluation

Matlab^®^ [72] was used for signal evaluation. The *V_RMS_* of the coupled signal and the noise *V_RMS_* were calculated. The *V_RMS_* was normalized for each forearm to eliminate inter-subject deviations, as described in Section 5.2.2.

The common-mode shielding and the active shielding group were compared in terms of their EMG *V_RMS_* and their noise *V_RMS_*. Chi-Square Goodness of Fit Test [73] and Bartlett Test [74] were used to test the groups for normal distribution and equal variances. The two-sample *t*-test [75] with the null hypothesis that the two groups are from populations with equal means was used with a significance level of 5%.

### 5.4. Proportionality to Force Level

To show the *V_RMS_* of the EMG signal at different force levels, Sensor 5 was applied above the flexor carpi radialis. Five able-bodied subjects aged between 26 and 52 were measured at the left and right forearm, which were treated as individual subjects in the data evaluation. A digital body scale was placed between thumb and the other digits and it was squeezed at the respective force level. The force levels were set in 10% steps of the MVC. The 5 s measurement with the oscilloscope [66] was started as soon as the respective force level was reached, so the onset of the muscle contraction was not included. Each subject performed the measurements once at each forearm and at each force level. The measurements were conducted in the order: 50%, 40%, 60%, 30%, 70%, 20%, 80%, 10%, 90%, 0%, and 100%. Resting periods of 40 s were added in between the measurements to avoid fatigue from influencing the measurements. The *V_RMS_* was normalized to eliminate inter-subject deviations.

### 5.5. Wearing Comfort

Five subjects answered a survey after the sensor comparison measurements, which comprised questions about the feel of the sensor when applying the measurement setup, the feel at the beginning and at the end of the measurements. It was asked if the subjects were sweating under the sensor or if pressure marks were obtained. The subjects were asked to rank the different sensor materials according to their preference for long-term use. In the survey, the sensors were categorized into copper, textile and flex sensors.

## 6. Results

### 6.1. Measured EMG Signal in the Time- and Frequency-Domains

Figure 13a shows a typical capacitively measured EMG signal with its corresponding amplitude spectrum (Figure 13b). The flex sensor (Sensor 5) and the electronics and software described above were used in this measurement.

### 6.2. Theoretical Analysis of the Measurement System

The entire sensor set-up was examined analytically and theoretical amplitudes were calculated for comparison with real-world measurements.

#### 6.2.1. Input Stage

In Figure 14, the calculated transfer functions of the input stage are plotted for the six different sensors. The equivalent circuit diagram and the transfer function are described in Section 4.2.

#### 6.2.2. Analog Bandpass and Digital Signal Processing

The transfer functions of the analog bandpass from Section 4.3 and the digital signal processing are plotted in Figure 15.

#### 6.2.3. Entire Measurement Set-Up

The transfer function of the entire measurement system (for plot see Figure 16a) was applied to the EMG input amplitude spectrum as defined in Figure 9 to obtain the results illustrated in Figure 16b. We computed the RMS values of the calculated amplitude spectra and normalized their sum to 1 for comparison to the real-world measurements (Section 6.3).

### 6.3. Comparison of Sensor Assemblies

The normalized mean RMS values and the standard deviations of the real-world measurements as well as the theoretical calculated RMS values are plotted in Figure 17. Since the Chi-Square Goodness of Fit Test [73] showed normal distribution, and the Bartlett Test [74] did not show equal variances, the two sample *t*-test [75] for equal means without the assumption of equal variances was applied. This *t*-test showed that the data resulted from different groups and that only Sensors 5 and 6, i.e., the sensor with the thin Platilon^®^ foil and the conductive sensor, fell within the same group; at a significance level of 5%, the *p*-value was 0.3. For the other group comparisons, the largest *p*-value was relatively small with $3.14\times 10^{-4}$; all other sensors therefore led to different coupled amplitudes.

Sensors 5 and 6 showed the highest coupled amplitudes. Sensor 6 does not have a dielectric covering the sensor area, while Sensor 5 has a very thin foil in front of the sensor area. At Sensor 5, the ratio of the coupling capacity to the parasitic capacity led to the high coupled amplitude. Sensor 4 is also a flex sensor, but—due to its capacity ratio—it exhibited the lowest amplitudes. Sensor 1 has the same dielectric covering the sensor area as Sensor 4, but the ratio of the dielectrics causes a higher signal amplitude. Sensors 2 and 3 have the same parasitic capacity, but Sensor 2 showed higher amplitudes because it has a higher coupling capacity.

#### 6.3.1. Fatigue Evaluation

The mean *V_RMS_normalized* did not decrease over the 18 measurements (see Figure 18). To resemble the sensor comparison measurements, the mean value of three subsequent measurements was calculated. No fatigue occurred in the sensor comparison measurements.

### 6.4. Shielding (Active Shield and Common-Mode Shield)

The normalized *V_RMS_* of the EMG signal and the noise values for both active and common-mode shielding are plotted in Figure 19. It can be seen that the amplitudes of the coupled EMG signal for active shielding (0.5056 *V_RMS_*) and that for common-mode shielding (0.4944 *V_RMS_*) are almost identical. The slight difference might be due to higher noise and lower parasitic capacity, but it is not statistically significant.

The mean of the normalized noise for active shielding (0.5845 *V_RMS_*) is slightly higher than that for common-mode shielding (0.4155 *V_RMS_*). The active shielding circuit feeds back both input signals via the voltage follower. These OpAmps introduce differential-mode noise, which is coupled directly to the input signal lines and further amplified.

The statistical evaluation shows that the values of all four bars are normally distributed. The active shield and the common-mode shield have equal variances in the comparison of the EMG signals and the noise. The null hypothesis of the two-sample *t*-test is that the data are from populations with equal means. For the EMG signal *V_RMS_*, the null hypothesis is not rejected (*p*-value: 0.0816). For the noise *V_RMS_*, the null hypothesis is rejected (*p*-value: $0.867\times 10^{-3}$), and therefore the noise is higher for active than for common-mode shielding at a significance level of 5%.

The EMG signal *V_RMS_* values of the two set-ups do not differ significantly, but the noise of the active shield set-up is higher, resulting in a lower SNR than for common-mode shielding. Furthermore, in the case of interferences, such as movement artifacts, the common-mode shield has a higher CMRR. For these reasons, the common-mode shielding was implemented in the final circuit of our low-noise high-stability sensor.

### 6.5. Proportionality to Force Level

The *V_RMS_normalized* is increasing with increasing muscle force level (see Figure 20). Muscle force level correlates with measured EMG *V_RMS_*. The resulting curve corresponds to force–EMG relations, as shown in the literature [76,77].

### 6.6. Power Consumption

The current of the sensor was measured with 6 mA at a supply voltage of 3.7 V (=22.2 mW). This includes the μC, which requires 2.7 mA. At this measurement, the controller was set to 16 MHz clock frequency and the ADC, DAC, brown out detector and IO-pins were activated. The current of the controller includes also the integrated OpAmps for gain adaption, the non-volatile memory and the digital signal processing software. The BLE module would increase the current at 3.3 V by 1 mA in advertising mode and by 10 mA in send/receive mode [78]. The send/receive mode is only used in the case of sensor configuration or data transfer to the PC.

### 6.7. Wearing Comfort

The survey showed that all sensors feel comfortable at the surface of the skin. The textile sensors (Sensors 2 and 3) have the highest wearing comfort, followed by the flex sensors (Sensors 4–6) and the copper sensor (Sensor 1) performed worst in the wearing comfort evaluation (Table 5).

### 6.8. Comparison of Capacitive and Conductive Sensors

The presented capacitive EMG sensor was compared with a conductive EMG sensor, the 13E401 from Otto Bock [79]. As described in Section 2, the conductive measurement principle results in higher signal coupling. The 13E401 has high signal quality when there is some sweat, which forms the electrolyte between the EMG sensor and the skin. However, it takes some time to form that sweat film. The impedance of the stratum corneum decreases in the course of time [80,81], therefore gain level adaptions are necessary, if it is desired to exploit full operating range. When there is poor contact, e.g., due to a missing sweat film or high hairiness, the signal has lower quality. The same applies to conventional EMG sensors, such as the SX230 from Biometrics Ltd. [82] or the Delsys Bagnoli [83]. Delsys defines the SNR of their EMG equipment to be 65 dB. These sensors have protruding metal parts to establish the conductive connection to the skin, which can cause pressure marks.

The MyoWare^TM^ EMG sensor [84] uses an electrolyte gel to establish the conductive connection to the skin. This sensor has high SNR straight away at application to the skin surface. However, gel sensors are not suitable for long-term application.

Although the conductive sensors have a high SNR at optimal skin contact, the presented capacitive sensor has a high SNR (40 dB at Sensor 1) as well and it is already given at application to the skin surface as it is independent of an electrolyte. The SNR would be even higher at sensor assemblies with better signal coupling (e.g., Sensor 5). Due to the flexible design, pressure marks can be avoided and especially the textile sensor feels comfortable to the skin (see Section 6.7).

### 6.9. Alternative Reference Design: INA with DC Rejection Reference Design

The current sensor circuit was also built with DC rejection reference design (see Section 3.2.2.3). In real-world applications, the circuit shows the desired filtering behavior directly at the first stage. However, when placing the sensor with asymmetric pressure, the signal at the INA and/or (depending on the interference) the feedback amplifier are/is saturated. The circuit is then unstable due to the feedback. Such instability is not acceptable in a real-world measurement system, so the INA with DC rejection reference design was not used in our final set-up. Further, fewer components are required when no DC rejection reference design is employed.

## 7. Discussion

### 7.1. Capacitive EMG Sensing Electrode

Six dry sensor set-ups were analyzed theoretically and measured to find the set-up with optimal signal coupling. Due to their flexible design, they adapt well to the forearm anatomy. The textile sensor in particular feels comfortable on the skin. A bio-compatible assembly is easy to achieve, since numerous bio-compatible insulating materials are available. This is, however, not the case for conductive materials. The textile and the copper sensors are connected to the electronics via coax-cables (UMCC), which have advantageous properties for signal transmission, but establishing a stable, conductive connection to the sensor area and the shielding is difficult. Mechanically, they do not exhibit the long-term stability of flex sensors, which are connected to a flat cable plug. To avoid short circuiting, insulating textile must not absorb sweat. Unlike textile or foil sensors, where sweat might travel along the UMCC cables into the sensors, flex sensors are insensitive to water. This problematic sensitivity to water could be addressed by sealing the cable inlets.

Due to the sensor design, the material forming the dielectric for the coupling capacity and also for the parasitic capacity must be flexible. For the coupling and parasitic capacities to remain constant, the material should further be incompressible and the dielectric constant should not change.

### 7.2. Body Reference

To shift the common-mode component of the measurement signal to within the operating range, we used a conductive reference in our set-up because it enables stable operation. Only one small reference is necessary for an entire array, and the reference is not as sensitive to lifting and changes in skin condition as in the case of a conductive sensor electrode. The reference can be placed at any part of the forearm where no movement is expected. A conductive reference is predominantly advantageous, and the advantage of the insulated measurement principle is maintained.

### 7.3. Analog Circuit

The bias resistor was chosen to be relatively low for stability, although—together with the parasitic capacity—it damps the signal. The damping effects due to parasitic capacities can be compensated by higher amplification.

### 7.4. Comparison of Sensor Assemblies

The measurements for the sensor comparison were conducted on healthy subjects and not on amputees; nevertheless, the results also extend to amputees. If the remaining muscle tissue is reduced, sensor area and inter-electrode distance and total gain can be adapted.

Differences between theory and real-world measurement may have resulted from the assumptions made for the calculation. The impedance of the stratum corneum was assumed to be an ohmic resistance, but its electrical properties vary depending on skin condition. Since the theoretical calculation is only an estimation of the insulated measurement, we consider the assumptions for the stratum corneum to be sufficient. The impedance in the measurements depends on the pressure applied to the sensor. Air entrapped between the sensor assembly layers may have led to differences in the capacities. Further, due to the sensor-making process, layer thickness may have differed from that given in the manufacturer data sheet.

Differences in the measurements may have resulted from slight variations in sensor positioning, although we strove for accurate placement. Further, the pressure of the cuff may have varied slightly between measurements. The subjects were asked to perform maximum voluntary contraction, so slight variations could not be excluded. These facts led to minor variations but the results are statistically sound. Although the sensors were placed above the musculus extensor digitorum, some of the EMG signal power might have resulted from other muscles due to crosstalk.

In terms of signal coupling, the conductive sensor (Sensor 6) had the highest amplitude in the real-world measurements. However, the *t*-test did not show a significant difference between the conductive sensor and Sensor 5, which has a thin foil as coupling dielectric. The capacitive sensor was more stable, as it is insensitive to skin conditions; adaption of the amplification for real-word application is therefore unnecessary. With increasing parasitic capacity, the common-mode rejection ratio increases, but the coupled signal amplitude is increasingly attenuated. By decreasing the parasitic capacity, the coupled signal amplitude can be increased. The capacity ratio can be set by selecting the appropriate dielectrics.

## 8. Conclusions

Insulated EMG measurement is associated with several challenges. Due to the capacitive measurement principle, the impedances at the coupling are much higher than in conductive measurement. Very small signals must be measured with precision electronics. Low-frequency movement artifacts pose a problem in biosignal measurement because they can have higher amplitudes than the EMG itself. Further, noise from various sources, such as the 50 Hz hum, disturbs the EMG signal.

Considering and addressing these challenges, we developed a highly stable insulated EMG measurement system. Our sensor is particularly suitable for the Otto Bock standard prosthesis in real-world applications. The set-up we have introduced has high signal quality (SNR = 40 dB at Sensor 1) and therefore allows accurate measurement. Using a sensor assembly with better signal coupling, such as Sensor 5, further increases the SNR. The prototype sensor is a compact two-layer PCB print with 27 mm × 46 mm and it requires only 22 mW. No feedback loops are included in our system to guarantee high stability. The flexible sensors avoid occurrence of pressure marks, and their design reduces movement artifacts. High wearing comfort is achieved, especially for the textile sensors, and skin irritations are avoided when using biocompatible insulating materials. The capacitive sensor shows greater stability because it is independent of skin condition, so no gain-level adaption is necessary.

Digital signal processing is essential to high-quality EMG measurement. We developed algorithms particularly for this system to achieve low power consumption and real-time capability for use in prosthesis.

As part of future work, we will improve the flexible sensors for higher mechanical stability. The textile sensors in particular are very promising due to their high wearing comfort, but they must be sealed to be waterproof; this is essential to long-term application, as otherwise sweat could cause short circuiting. The sensor can be washed when waterproof, which is a requirement in prosthesis applications.

Further, we will ask amputees to test our sensor set-up to identify potential challenges associated with its use in prostheses. The insulated EMG sensor can then be adapted to enable even more stable operation.

Possible future applications of this sensor include, for example, exoskeletons, sport physiology, medical diagnosis and analysis of human movement by means of EMG signals in real-world environments.

## Figures and Tables

**Figure 1 sensors-19-00961-f001:**
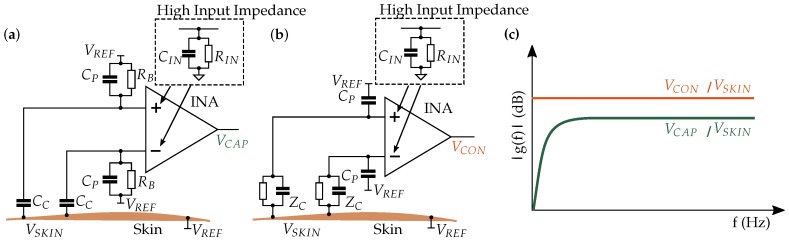
Comparison of the capacitive and conductive measurement principles. (**a**) Capacitive measurement: The coupling capacity *C_C_* forms a voltage divider with the parasitic capacity *C_P_* and the bias resistor *R_B_*, which is necessary for the bias current. (**b**) Conductive measurement: The high input impedance of the amplifiers can be exploited, because no bias resistor is required. *Z_C_* is the coupling impedance. (**c**) Transfer functions of the measurement principles.

**Figure 2 sensors-19-00961-f002:**
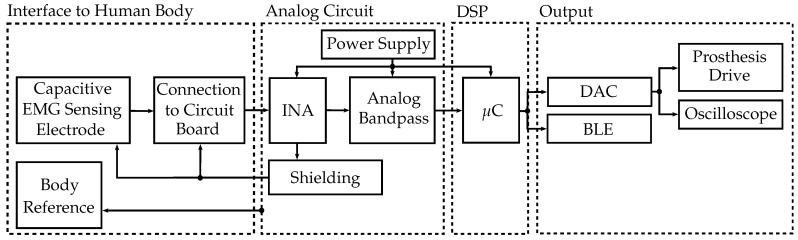
Block diagram outlining the insulated EMG measurement system. The capacitive EMG sensing electrode is connected to the analog circuit, where it is preamplified and filtered. The analog circuit provides shielding and a body reference. Digital signal processing (DSP) is performed in the microcontroller (μC). The digital-to-analog converter (DAC) is connected to the prosthesis drive in real-world applications. Bluetooth-low-energy (BLE) and an oscilloscope were used in experiments.

**Figure 3 sensors-19-00961-f003:**
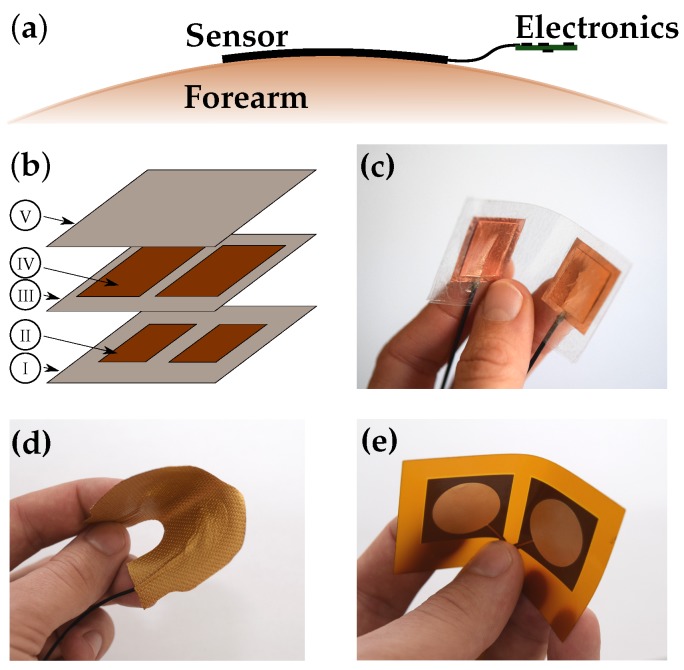
(**a**) Adaptation of the flexible sensor to the human forearm anatomy; (**b**) structure of the multilayer construct ((I) dielectric 1; (II) sensor areas; (III) dielectric 3; (IV) shielding; and (V) dielectric); (**c**) copper sensor (Sensor 1); (**d**) textile sensor (Sensors 2 and 3); and (**e**) flex sensor (Sensors 4–6).

**Figure 4 sensors-19-00961-f004:**
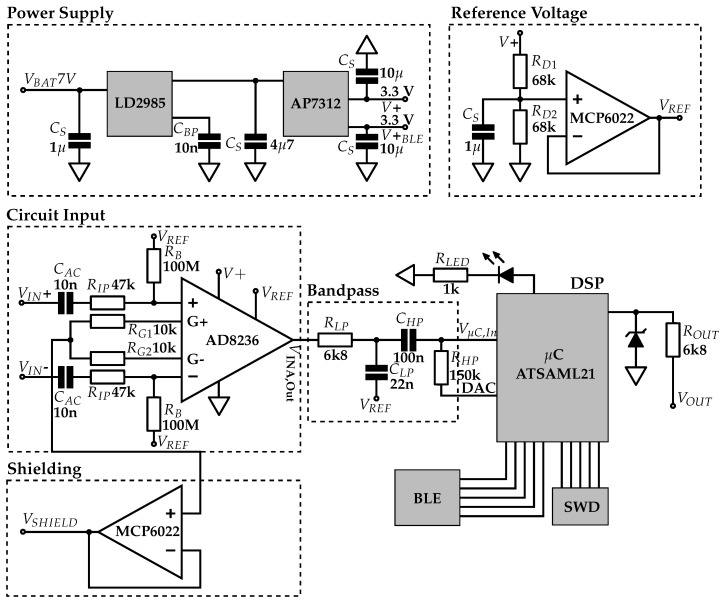
Circuit diagram of insulated EMG measurement sensor.

**Figure 5 sensors-19-00961-f005:**
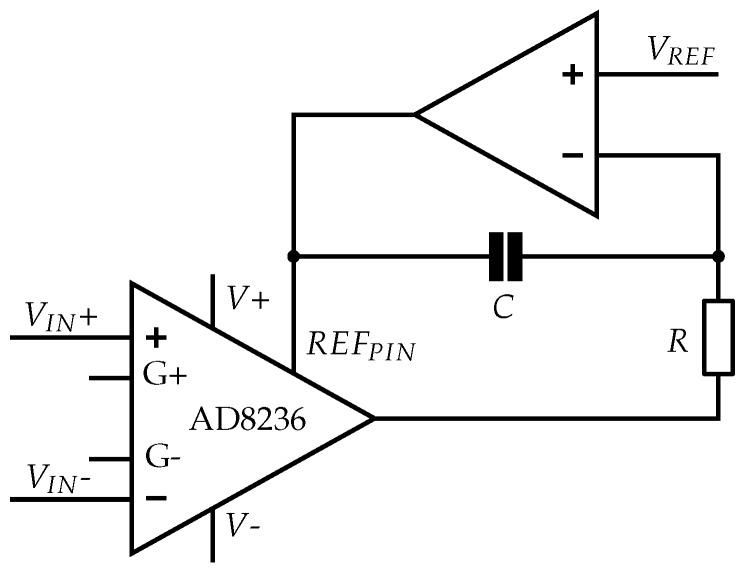
INA with DC rejection reference design with negative feedback of the low-frequency signals to the *REF_PIN_* of the AD8236 instrumentation amplifier. This circuit leads to a highpass with cutoff frequency *f_C_* (23.4 Hz) defined by *R* (68 kΩ) and *C* (100 nF) (see Equation (Equation 2)) [42].

**Figure 6 sensors-19-00961-f006:**
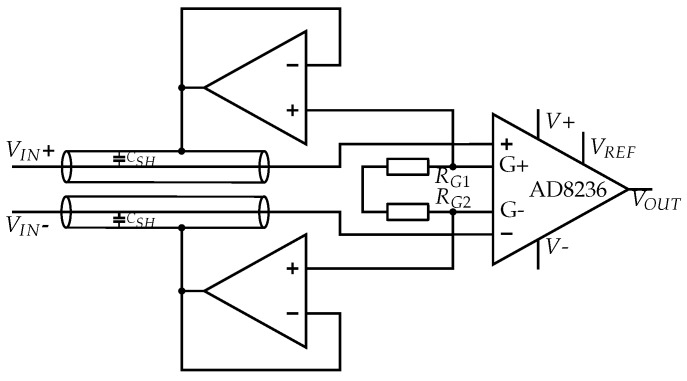
A separate active shield for each signal line decreases parasitic capacities and protects against external interferences.

**Figure 7 sensors-19-00961-f007:**
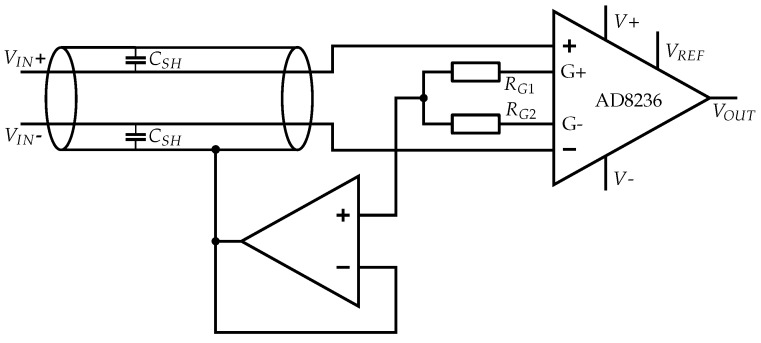
Common-mode shield for increasing the common-mode rejection ratio and protecting against external interferences.

**Figure 8 sensors-19-00961-f008:**
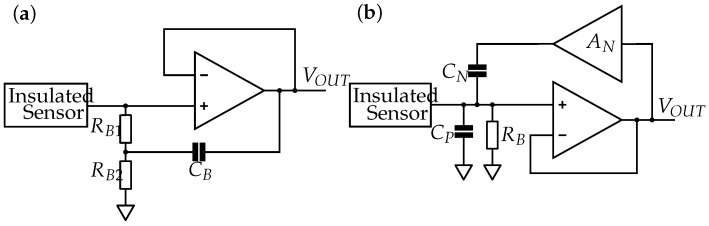
(**a**) Bootstrapping via $C_{B}$ to increase the input impedance while maintaining a DC path for the bias current. (**b**) Neutralization to reduce parasitic capacity *C_P_*. *A_N_* is the gain to be set for the feedback loop via *C_N_*. Adapted from [56].

**Figure 9 sensors-19-00961-f009:**
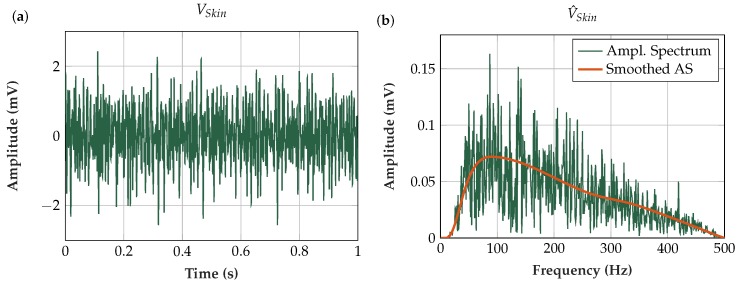
(**a**) Theoretical input signal *V_Skin_* in time domain (duration = 1 s, $f_{S}$ = 10 kHz, peak to peak amplitude = 5 mV). (**b**) Amplitude spectrum of the theoretical input signal (resolution Δf = 1 Hz).

**Figure 10 sensors-19-00961-f010:**
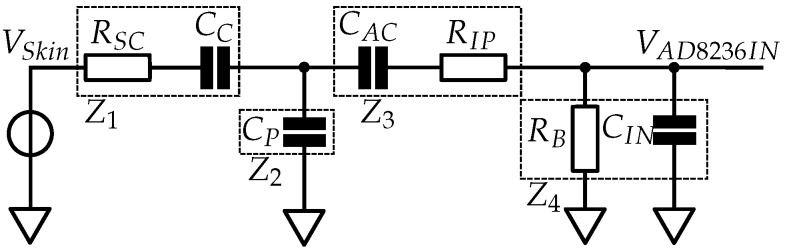
Equivalent circuit diagram of the input stage used for the transfer function to calculate the coupled signal amplitude.

**Figure 11 sensors-19-00961-f011:**
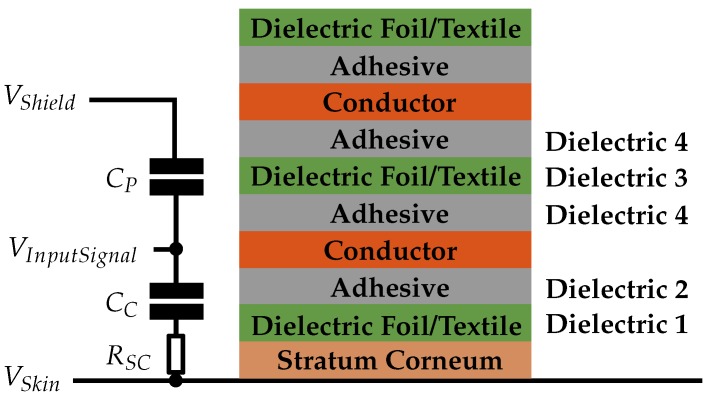
Stacking of the sensor assemblies in detail. The adhesives are silicone or acrylic adhesive and are not included in all sensor assemblies, as described in Section 3.1.1. The parameters of the dielectrics are listed in Table 2.

**Figure 12 sensors-19-00961-f012:**
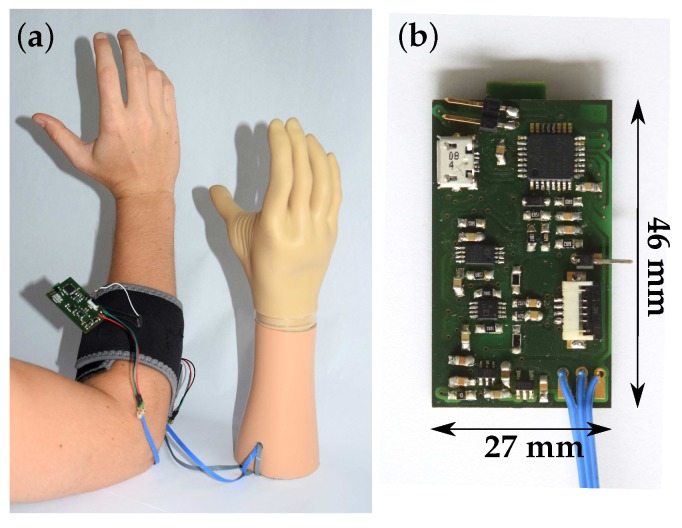
(**a**) Set-up of the capacitive EMG measurement system. EMG sensor fixed to a human forearm by a cuff. Control of myoelectric handprosthesis. (**b**) Front side of the PCB Print.

**Figure 13 sensors-19-00961-f013:**
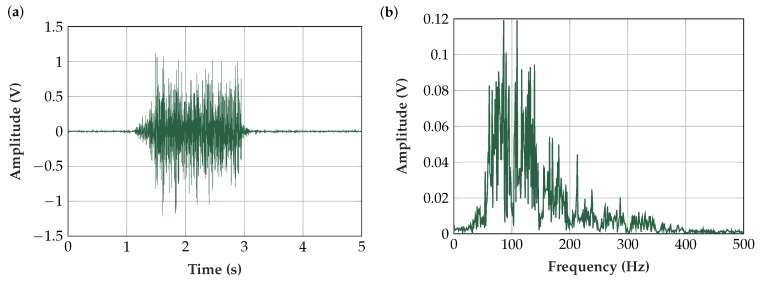
(**a**) EMG contraction measured with our insulated EMG measurement set-up using Sensor 5 (flex sensor with Platilon^®^ foil) and an oscilloscope [66] at 10 kHz sampling frequency; and (**b**) amplitude spectrum of contraction EMG at 1.7–2.7 s (resolution Δf = 1 Hz). The damping characteristic of the highpass filters and the notch filter at 50 Hz and its harmonics can be seen.

**Figure 14 sensors-19-00961-f014:**
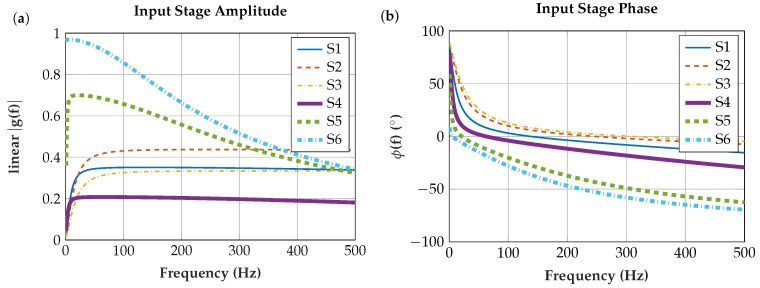
Transfer functions from input stage to input of the INA for the six different sensor assemblies: (**a**) amplitude response; and (**b**) phase response.

**Figure 15 sensors-19-00961-f015:**
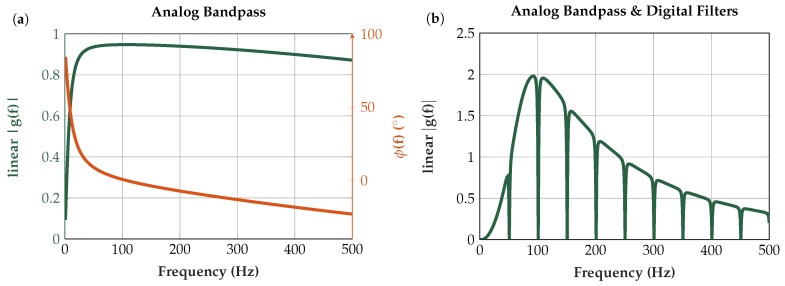
(**a**) Transfer function of the first-order analog bandpass ($f_{C}$: 11–1064 Hz); and (**b**) combined transfer function of analog bandpass and digital signal processing. The digital signal processing includes a second-order highpass with 60 Hz cutoff frequency and a second-order notch filtering 50 Hz and its harmonics. A lowpass follows after the highpass and notch to eliminate high-frequency noise [53].

**Figure 16 sensors-19-00961-f016:**
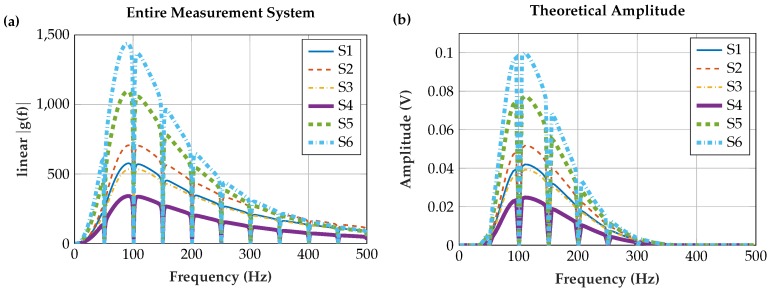
(**a**) Transfer function of the entire measurement system. Cascading of the transfer functions of the input stage, analog bandpass, digital signal processing and frequency independent amplification. (**b**) Calculated EMG amplitude spectrum (resolution Δf = 1 Hz). The transfer function of the entire measurement system in (**a**) was applied to the theoretical EMG input amplitude spectrum as defined in Figure 9. The resulting shape is in accordance with the measured EMG amplitude spectrum in Figure 13, as the theoretical and the real measurement system have similar transmission behavior. Note that the theoretical amplitude is based on the smoothed amplitude spectrum.

**Figure 17 sensors-19-00961-f017:**
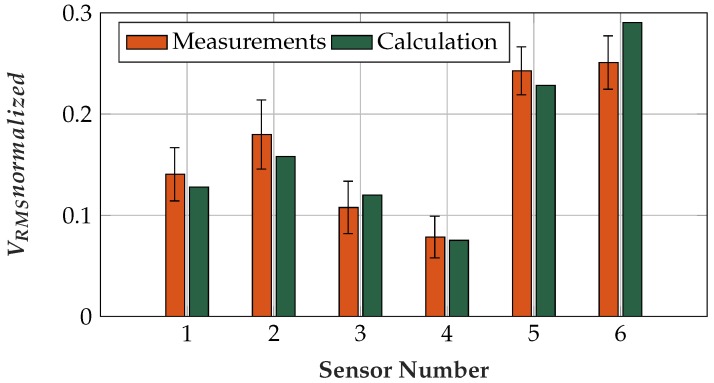
Normalized measured mean *V_RMS_* values (including standard deviations) and normalized calculated *V_RMS_* values for comparing the coupling characteristics of different sensor assemblies.

**Figure 18 sensors-19-00961-f018:**
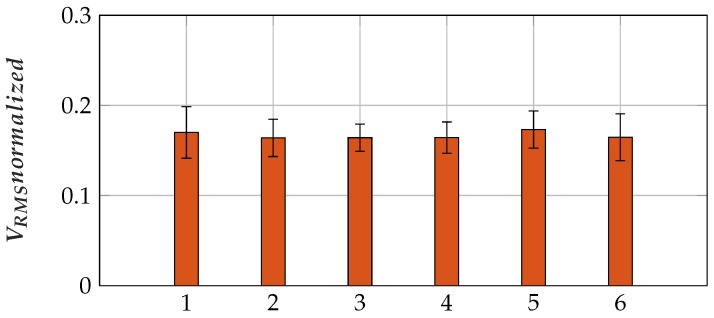
Normalized *V_RMS_* values did not decrease over the 18 maximum voluntary contractions measured with Sensor 5. One bar comprises the mean of three subsequent measurements of five subjects.

**Figure 19 sensors-19-00961-f019:**
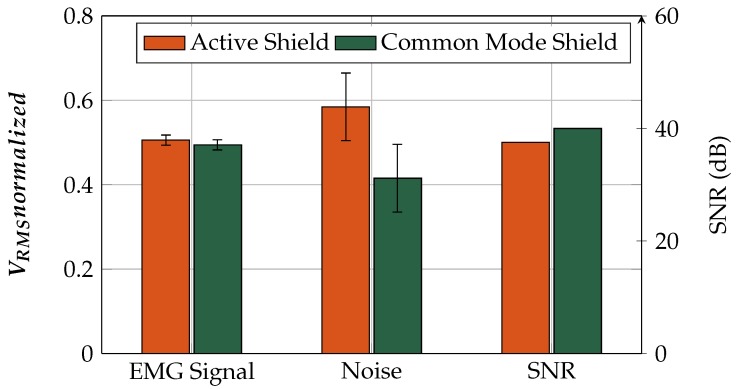
Normalized *V_RMS_* of EMG signal, and noise for active and common-mode shielding. Note that the *V_RMS_* values are normalized, so the sum of the mean *V_RMS_* at each comparison results to 1. The absolute EMG *V_RMS_* are considerably higher than the noise *V_RMS_* values. SNRs of the absolute mean EMG *V_RMS_* and the absolute mean noise *V_RMS_* are shown.

**Figure 20 sensors-19-00961-f020:**
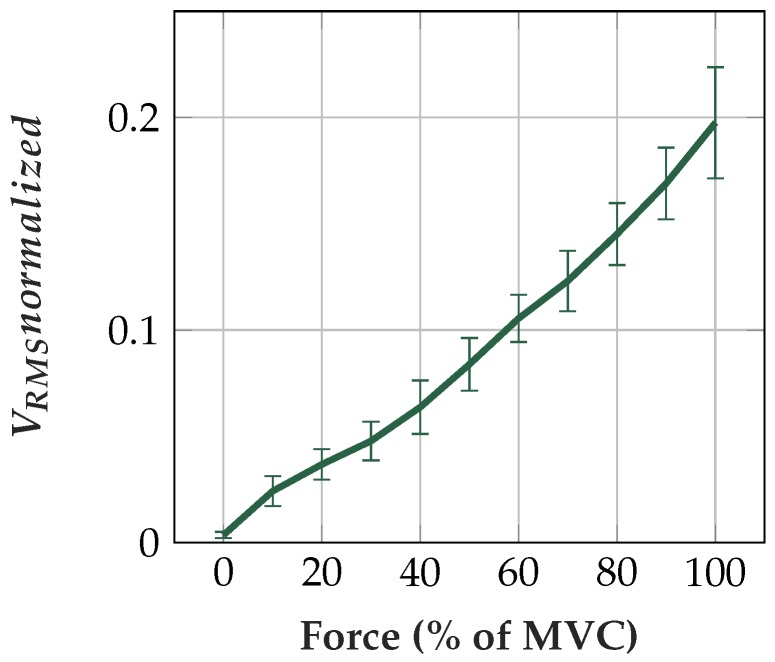
Normalized *V_RMS_* at different muscle force levels.

**Table 1 sensors-19-00961-t001:** Parameters of sensor assemblies.

Sensor #	Sensor 1	Sensor 2	Sensor 3	Sensor 4	Sensor 5	Sensor 6
Material	Copper & Insulation foils	Textiles	Textiles	Flex print	Flex print	Flex print
Sensor area shape	rectangular	circular	circular	circular	circular	circular
Width/radius (mm)	15	7.5	7.5	10	10	10
Height (mm)	16.5	-	-	-	-	-
Area (mm^2^)	247.5	176.7	176.7	314.2	314.2	314.2
Inter-electrode distance (mm) (center to center)	35	32.5	32.5	30	30	30
Adhesive (Dielectric 2)	Acrylic adhesive [26]	Silicone	Silicone	Acrylic adhesive [26]	Acrylic adhesive [26]	Acrylic adhesive [26]
Ad. Thickness (μm)	2*30	100	100	30	neglected [27]	neglected [27]
Ad. $\varepsilon_{R}$	3.8 [26]	2.8 [28]	2.8 [28]	3.8 [26]	-	-
Dielectric 1 (between skin and sensor area)	self-adhesive PVC foil [29]	Exoflex [30] (TPU)	PU [31]	self-adhesive PVC foil [29]	Platilon^®^ foil [32]	-
Diel. 1 thickness (μm)	90	75	250	90	25	-
Diel. 1 $\varepsilon_{R}$	3.19 [33]	6.6 [34,35]	6.6 [34,35]	3.19 [33]	6.6 [34,35]	-
Adhesive (Dielectric 4)	Acrylic adhesive [26]	Silicone	Silicone	Acrylic adhesive [26]	Acrylic adhesive [26]	Acrylic adhesive [26]
Ad. Thickness (μm)	30	100	100	neglected [27]	neglected [27]	neglected [27]
Ad. $\varepsilon_{R}$	3.8 [26]	2.8 [28]	2.8 [28]	-	-	-
Dielectric 3 (btw. sensor area and shield)	Tesafilm ^TM^ [36]	Exoflex [30] (TPU)	Exoflex [30] (TPU)	Polymide [27]	Polymide [27]	Polymide [27]
Diel. 3 thickness (μm)	65	75	75	50	50	50
Diel. 3 $\varepsilon_{R}$	3.19 [33]	6.6 [34,35]	6.6 [34,35]	3.4 [27]	3.4 [27]	3.4 [27]

**Table 2 sensors-19-00961-t002:** Calculated coupling capacity *C_C_* of various sensor assemblies. Dielectric 1 is the dielectric foil or textile. Dielectric 2 is the silicone or acrylic adhesive.

#	Diel. 1 Thickness (μm)	ε _R,1_	Diel. 2 Thickness (μm)	ε _R,2_	Area (mm^2^)	C_C_ (pF)
1	90	3.2	60	3.8	247	49.7
2	75	6.6	100	2.8	177	33.3
3	250	6.6	100	2.8	177	21.3
4	90	3.2	30	3.8	314	77.0
5	25	6.6	-	-	314	733.6
6	-	-	-	-	-	-

**Table 3 sensors-19-00961-t003:** Shield of connection to circuit board [25,27,65].

Connection	Length *l* (mm)	Width *w* (mm)	Thickness *d* (μm)	$r_{\mathrm{outer}}$ (μm)	$r_{\mathrm{inner}}$ (μm)	ε _R_	$C_{P,\mathrm{Connection}}$ (pF)
UMCC	40	-	-	100	83	2	2
Flex print	70	50	1	-	-	3.4	42

**Table 4 sensors-19-00961-t004:** Calculated parasitic capacity *C_P_* of various sensor assemblies. Dielectric 3 is the dielectric foil or textile. Dielectric 4 is the silicone or acrylic adhesive. *C_P_* includes the capacity of the connection in Table 3.

#	Diel. 3 Thickness (μm)	ε _R,3_	Diel. 4 Thickness (μm)	ε _R,4_	Area *A* (mm^2^)	*C_P_* (pF)
1	65	3.2	30	3.8	271.7	87.2
2	75	6.6	100	2.8	194.7	38.7
3	75	6.6	100	2.8	194.7	38.7
4	50	3.4	-	-	408.2	287.8
5	50	3.4	-	-	408.2	287.8
6	50	3.4	-	-	408.2	287.8

**Table 5 sensors-19-00961-t005:** Wearing comfort of different sensor materials.

Sensor	Mean Comfort	Mean Rank
	(1 …very good, 5 …very bad)	(1 …best, 3 …worst)
Copper Sensor	2.80	3.00
Textile Sensor	1.24	1.20
Flex Sensor	1.64	1.80

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
