# Peer review of "An Insulated Flexible Sensor for Stable Electromyography Detection: Application to Prosthesis Control"

_sensors, 2019, doi:10.3390/s19040961_

Round 1

Reviewer 1 Report

The manuscript “An Insulated Flexible Sensor for Stable Electromyography Detection: Application to
Prosthesis Control” designed capacitive sensors for EMG measurement. The authors did a detailed theoretical analysis to prove the characteristics of the sensor. The major concern from this author is that it lacked the analysis with the real EMG data. The comments are as follows:

1.      The analysis with the input of real EMG data is needed. This study did a lot of theoretical analysis, however, it would be more convincible presenting the results with the real EMG signal, instead of simulated EMG data.

2.      As EMG is force sensitive, the measurement of EMG signals with the increase of force (muscle contraction) should be presented.

3.      It was mentioned that the proposed sensor had advantages over the conventional EMG sensors. It would be better if the authors could present a comparison with some commercial EMG sensors, such as delsys or biometrics, to support the point.

Author Response

Thank you for your appreciation of our work and thank you for your comments, which helped us with improving our work. Please find a point-by-point response to your comments in the Word file.

Reviewer 2 Report

According to the authors, the purpose of the study was to investigate the effectiveness of various capacitive sensors set-ups (6 in total) that did not require conductive connection to the skin, electrode gel, or skin preparation. In addition, the flexible sensors arrangements were reported to be: 1) stable; 2) have low power consumption; 3) low cost; 4) small; 5) easy to use; 6) comfortable; 7) real time capability; and 8) have a high signal to noise ratio. Furthermore, the sensors were implied to be able to enable prosthetic control of highly dexterous movements. Finally, the authors endeavored to find which of the sensor assemblies had the most optimal signal coupling. A large number of theoretical analyses, but limited measurements in human subjects were used to demonstrate that a capacitive sensor developed by the authors met the above criteria. In regards to the experiments in human subjects, the 6 sensors assemblies were worn during 3 maximum contractions of the extensor digitorum for each arm (36 total contractions) in 10 subjects who displayed a large age range (23-62). In a smaller experiment, 4 subjects were used in similar circumstances to compare shielding circuits. Apparently, the main findings were that assemblies 5 and 6 showed the highest coupling amplitudes.

Overall, the manuscript is well-written, and appears to have very few grammatical or typographical errors (although a few are present). It also attempts to extend previous research by other research groups. In addition, most of the theoretical data appear to have been computed carefully, although the measurements in human subjects were not described very well (see below). The sensor assemblies seem to be novel for the application described although different applications of similar assemblies have been used before. The topic overall is important to researchers and clinicians in rehabilitation fields. The focus of the research seems to be appropriate for Sensors and of interest to many readers of the journal.

However, I have several basic issues that should be addressed for this study to be publishable, mainly in regards to the experiments done on human subjects. Major issues are described in numbers 1-3 below and a few minor issues follow in 4-5.

1. In general, one major issue throughout the paper is that some of the assertions of the authors in regards to the characteristics of the sensors (characteristics that were alluded to throughout the paper) in lines 123-131 were not backed by any real experimental data. While some may have been demonstrated in the theoretical portion of the paper or are somewhat self-evident (e.g. cost, although differences in cost were not given), others had no data backing by data at all. As one of a few possible examples, it was implied throughout the paper that the sensors, or some of them, were very comfortable compared to traditional sensors or to the various sensors in the paper. However, it appears that no data was reported from the human subjects to support this assertion. There were no questionnaires or surveys given to the participants asking them to rank or subjectively report the comfort of the 6 assemblies. Furthermore, I assume that the subjects were not blinded to which sensor they were wearing during the experiments. It is also unclear if the subjects were naïve to the purpose of the experiment or the assumed characteristics of the sensor assemblies.

2. Another major potential issue is the methodology used in the experiment on the 10 subjects. First, if the investigators where interested in the forearm musculature why did they only measure from the extensor digitorum? Why not ECRL, ECRB, ECU or all of them? No antagonist muscles were analyzed despite referring to their importance for prosthetic control in the Introduction. Second, why were only MVCs analyzed? In the Introduction the authors explain the importance of prosthetic control for dexterious manipulation and real world application. However, dexterious manipulation is almost always done at low forces with low EMG levels and a large portion of this is in non-isometric (anisometric) conditions. Wouldn't it be better to at least have used low isometric forces and EMG levels? Some problems in interpreting the EMG signal vary with force level. Accordingly, amplitude cancellation of the EMG signal (Day and Hulliger, 2001; Keenan et al 2005; Keenan et al 2006), which distorts the onsets of activation of muscles at low forces (Jesunathadas et al 2012) is an issue that could influence prosthetic control. Fourth, the execution and quantification of the MVCs were not described at all. Was rate of force production controlled? How long of a contraction time was used for analysis? How was the contraction performed? Against a restraint? How much rest between the large numbers (36) of MVCS? Was rest period monitored?

3. Starting about line 634, the authors describe how fatigue may have occurred but assure the reader that it did not influence the stats due to randomization. Why was fatigue not quantified simply by measuring MVC force or even maximum EMG over the course of the 36 contractions? This is the only way, but an extremely easy way to quantify fatigue. If there was no fatigue then MVC force should remain statistically similar across the 36 trials and not decline. Furthermore, if maximum EMG remained similar that would be some additional evidence. It appears that the authors neglected to quantify these variables or did not report them. Randomization alone appears to be not enough as MVC force could have been reduced in general across all condition orders with time.

4. How could volume conducted signals (crosstalk) have influenced the results? This is one of several issues that probably should have at least been touched on in the Discussion.

5. In a more minor point, the authors do not do a very good job of summarizing the main findings at the end of the abstract or the Discussion in an easy to understand and concise manner.

Author Response

Response to Reviewer 2 Comments

Thank you for your appreciation of our work and thank you for your comments, which helped us with improving our work. Please find below a point-by-point response to your comments.

Point 1: In general, one major issue throughout the paper is that some of the assertions of the authors in regards to the characteristics of the sensors (characteristics that were alluded to throughout the paper) in lines 123-131 were not backed by any real experimental data. While some may have been demonstrated in the theoretical portion of the paper or are somewhat self-evident (e.g. cost, although differences in cost were not given), others had no data backing by data at all. As one of a few possible examples, it was implied throughout the paper that the sensors, or some of them, were very comfortable compared to traditional sensors or to the various sensors in the paper. However, it appears that no data was reported from the human subjects to support this assertion. There were no questionnaires or surveys given to the participants asking them to rank or subjectively report the comfort of the 6 assemblies. Furthermore, I assume that the subjects were not blinded to which sensor they were wearing during the experiments. It is also unclear if the subjects were naïve to the purpose of the experiment or the assumed characteristics of the sensor assemblies.

Response 1: Quantized results of the defined requirements were added to the paper (SNR, power consumption, signal delay and real-time capability, comfort, compact size). Differences in cost were not given, as it would be misleading to compare a prototype with a commercial EMG sensor that had to undergo medical certification. The survey asking patients about comfort of the different sensor assemblies was added to the paper to support this assertion. The patients were not blinded physically to which sensor they were wearing, however, they were naïve to the purpose of the experiments and they didn’t have the theoretical background. This information was added to the paper.

Point 2: Another major potential issue is the methodology used in the experiment on the 10 subjects. First, if the investigators where interested in the forearm musculature why did they only measure from the extensor digitorum? Why not ECRL, ECRB, ECU or all of them? No antagonist muscles were analyzed despite referring to their importance for prosthetic control in the Introduction. Second, why were only MVCs analyzed? In the Introduction the authors explain the importance of prosthetic control for dexterious manipulation and real world application. However, dexterious manipulation is almost always done at low forces with low EMG levels and a large portion of this is in non-isometric (anisometric) conditions. Wouldn't it be better to at least have used low isometric forces and EMG levels? Some problems in interpreting the EMG signal vary with force level. Accordingly, amplitude cancellation of the EMG signal (Day and Hulliger, 2001; Keenan et al 2005; Keenan et al 2006), which distorts the onsets of activation of muscles at low forces (Jesunathadas et al 2012) is an issue that could influence prosthetic control. Fourth, the execution and quantification of the MVCs were not described at all. Was rate of force production controlled? How long of a contraction time was used for analysis? How was the contraction performed? Against a restraint? How much rest between the large numbers (36) of MVCS? Was rest period monitored?

Response 2: When comparing different sensor assemblies, we decided to use only one placement to avoid many factors from influencing the results. The signal coupling of the sensor, but not the different muscle groups should be compared. Measurements showing the relation between muscle force and EMG, measured at the flexor carpi radialis, were added to the paper. This way presenting the functioning of the sensors at low forces. The onset of the EMG contraction was not evaluated as the first two seconds of the sensor comparison measurements were discarded. This was described in more detail and the relevant references were included to the paper. The information regarding the measurement (execution, quantification, force production, contraction, restraint, rest period) was added to the paper. The consistency of the EMG throughout the 18 measurements was investigated in the fatigue evaluation. Please note that 36 measurements were performed per subject, and 18 measurements per forearm (left and right). This was written in a clearer way in the paper.

Point 3: Starting about line 634, the authors describe how fatigue may have occurred but assure the reader that it did not influence the stats due to randomization. Why was fatigue not quantified simply by measuring MVC force or even maximum EMG over the course of the 36 contractions? This is the only way, but an extremely easy way to quantify fatigue. If there was no fatigue then MVC force should remain statistically similar across the 36 trials and not decline. Furthermore, if maximum EMG remained similar that would be some additional evidence. It appears that the authors neglected to quantify these variables or did not report them. Randomization alone appears to be not enough as MVC force could have been reduced in general across all condition orders with time.

Response 3: The quantification of fatigue by measuring the RMS of EMG over the 18 measurements was added. It was shown, that the RMS does not decline over the 18 measurements. The information about resting periods was added to the paper.

Point 4: How could volume conducted signals (crosstalk) have influenced the results? This is one of several issues that probably should have at least been touched on in the Discussion.

Response 4: We added the issue of crosstalk to the discussion as suggested.

Point 5: In a more minor point, the authors do not do a very good job of summarizing the main findings at the end of the abstract or the Discussion in an easy to understand and concise manner.

Response 5: Concise and quantitative findings were added to the abstract and the conclusion of the revised manuscript.

Round 2

Reviewer 1 Report

The authors addressed all the points of the first round.

Reviewer 2 Report

The authors seem to have addressed all of my comments. I have no further concerns.